# FaceZip: Automatic Texture Compression for Facial Blendshapes

Cai Yiyi*
The University of Tokyo

Shinichi Kinuwaki†
Unaffiliated

Nobuyuki Umetani‡
The University of Tokyo

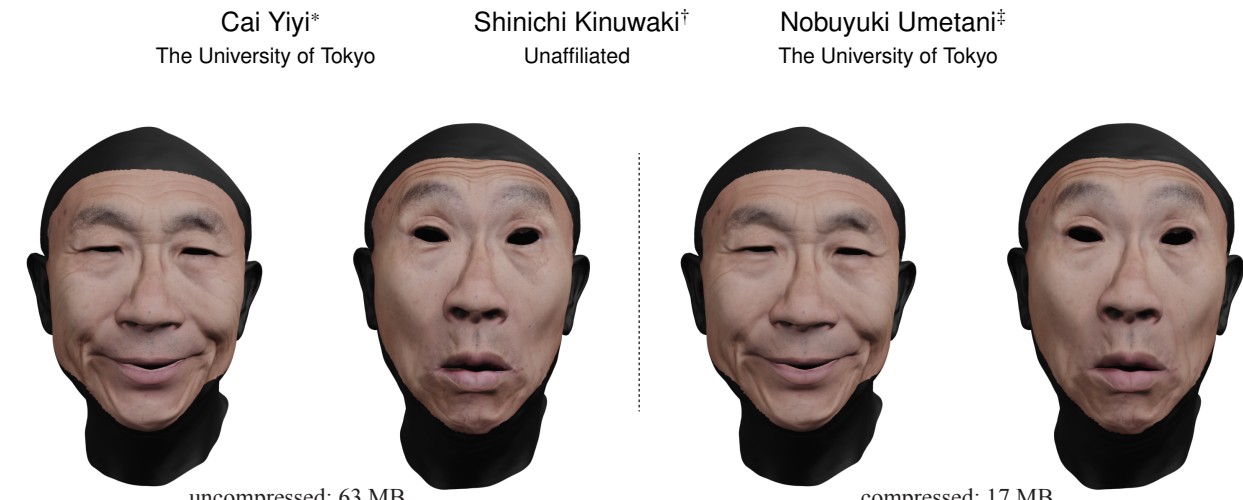

Figure 1: We present a method to compress the information of the texture that dynamically changes with the underlying blendshape mesh. Compared to the naïve method of storing all the textures on the blendshape mesh (left), our approach (right) significantly reduces the size with only a small loss of quality.

## ABSTRACT

Recently, numerous cinematic and interactive entertainment production companies have adopted advanced capture systems to acquire faithful facial geometries and corresponding textures. However, it is difficult to animate these models in a controllable manner for real-time applications. Although blendshape is typically used for parameterizing facial geometries, dynamically changing the texture of the geometry is challenging. Because texture data are significantly larger than the vertex coordinates of the meshes, storing the texture of all the blendshape meshes is impractical. We present a method to compress texture data in a manner compatible with blendshape for real-time applications such as video games. Our method takes advantage of the locality of facial texture differences by blending a few textures with spatially different weights. Our method achieved a more accurate reconstruction of the original textures compared with the baseline principal component analysis.

**Index Terms:** Computing methodologies—Computer graphics—Image Compression; Computing methodologies—Computer graphics—Texturing

## 1 INTRODUCTION

Recently, the demand has significantly increased for realistic digital human models in various applications such as cinema, interactive entertainment, and metaverse photogrammetry, which are typically used to automatically capture the 3D geometries and materials of actors. However, a wide range of skilled artists in modeling, sculpting, texture painting, rigging, and animation are still necessary to bring life to the captured model. This difficulty largely originates from the parameterization of the acquired model to efficiently represent the surface material and deformation in an efficient and controllable

*e-mail: caiyiyi1998@gmail.com
†e-mail: shinichikinuwaki@gmail.com
‡e-mail: n.umetani@gmail.com

manner. The overall facial deformation is typically parameterized using a blendshape model, and the detailed surface appearance is represented by 2D textures. However, their coupling, which dynamically changes the texture according to the blendshape parameter, has been challenging. It is difficult to represent details such as wrinkles, dimples, and furrows, which are dynamically created using various expressions.

Blendshape is a popular facial animation technique because it can create complicated deformations using a simple linear combination of semantically meaningful (typically 50–100) blendshape meshes. Theoretically, the texture of the blendshape can also be represented by a linear combination of the textures of the blendshape meshes. However, storing all textures of the blendshape mesh is not practical, as the memory requirement of the texture is significantly larger than that of the vertex coordinates of the blendshape mesh. Particularly in real-time applications such as games, the current hardware requires keeping the number of texture accesses at 10 or fewer.

A typical technique used in the gaming industry is to store the texture of a neutral face and several textures that collect wrinkles from large expressions (e.g., [26]). These textures are blended *nonuniformly* using the weight map, that is, spatially changing weights. The advantage of this approach is that the artist has control over all the textures, thereby allowing exaggerated facial expressions to be authored. However, this is time-consuming and requires a great deal of expertise to create such weight maps and wrinkle textures manually.

We present a technique for automatically generating a compressed texture for a blendshape model (see Figure 1). Our algorithm leverages the locality of wrinkles by dividing the entire texture into several fragments. These fragments were seamlessly stitched back to several textures while avoiding blur using our optimized selection of fragment combinations. Furthermore, we extend our model to the blendshape generated from the example-based blendshape technique [13], which is a popular method for generating blendshape meshes of Ekman's Facial Action Coding System (FACS) [6] from fewer example meshes.

We demonstrate our approach by comparing it against the baseline of principal component analysis (PCA). The contributions of the proposed method include the following:

- Automatic compression of textures for the application of facial blendshape.

- The extension of the texture generation for the blendshape generated by the example-based blendshape technique [13].

## 2 RELATED WORK

The generation of facial models has been studied extensively for several years. We refer the reader to surveys [18, 29] for a comprehensive review. This study focuses on texture compression of blendshape in a real-time environment.

**Parametric Facial Deformation** Skinning [11] is one of the simplest methods of animating faces by placing fictitious facial bones under the skin. Because it is difficult to faithfully reproduce facial deformation using bones, blendshape deformation [20] is often used instead of high-quality animation. The blendshape typically deforms the vertices of the meshes with a linear combination of differences from the neutral mesh [12]. Neumann et al. [19] proposed a method that extracts sparse and localized deformation modes from an animated mesh sequence, such that the extracted dimensions often have interpretable meanings. These parameterizations work well for the vertices of meshes, but often have difficulty handling textures because the amount of data is large.

**Expression Capture** High-quality facial capture setups are becoming increasingly common in industry. These systems typically use the photometric stereo technique with polarizers to obtain the albedo, specular, normal, and rough textures from vertical and parallel polarization images [8]. Riviere et al. [22] avoided sequential flashing of light when estimating these textures by applying inverse rendering to cross- and parallel-polarized images. Zhang et al. [28] modified the Light Stage [4] for high-speed cameras and developed a mechanism to directly capture the animation sequence itself at the video rate, instead of discrete expressions. Whereas expression capture requires multiple separate workflows, such as

mesh reconstruction, fitting to a base mesh, and the computation of each texture, Liu et al. [15] proposed a single end-to-end neural network acquisition framework.

**FACS Blendshape Generation** Productions typically use the meshes of FACS poses to create the blendshapes for facial animation. Because FACS is typically composed of 50 to over 100 independent meshes, there is a demand to build all FACS blendshape meshes from a limited number of captures. Li et al. [13] presented a retargeting technique for synthesizing blendshape meshes of FACS poses from a small number of expression captures. However, that study focused on facial geometry rather than texture. Li et al. [14] proposed a neural network for generating FACS expressions and their corresponding textures from a single neural facial scan. However, the resulting textures are too expensive to blend directly in real time.

**Facial Material Representation** Facial animation has been studied for many years. However, few studies have focused on the compression of facial textures. The 3D morphable model (3DMM) represents the detailed 3D model in lower dimensional parameters. We refer to a recent survey on morphable facial models by Egger et al. [5]. These parametric models can also decompose animation data and change facial expressions by changing the parameters. The pioneering work of Blanz et al. [3] parameterized both vertex coordinates and RGB texture values using PCA.

Recent studies using neural networks are expensive to evaluate in real time for high-resolution material generation. Machine learning approaches use convolutional neural networks (CNN) [21, 24] to directly output meshes and textures. But they all need a large amount of training data and cannot be used for real-time purposes. Lombardi et al. [16] represented facial data as a set of neural radiance field models, instead of a mesh and texture. The performance of the learning method can be extremely poor if the real data are significantly different from the training data in terms of age, skin color, etc.

Garrido1 et al. [7] and Shi et al. [25] presented small surface details using shape-from-shading techniques that dynamically generate a highly detailed surface geometry according to the expressions. However, these approaches have extremely high computational complexities and are unsuitable for real-time applications. Huang et al. [9] and Ma et al. [17] generated the details of different expressions using detailed maps while keeping the diffuse texture unchanged. However, a diffuse map typically changes as wrinkles appear, depending on the expression used.

## 3 METHODS

**Texture Compression** Let us assume we have $N$ number of RGB textures $\mathcal{I}_n$ where $n \in \{1, ..., N\}$ and one RGB "neutral texture" $\mathcal{I}_0$. Our algorithm computes four RGB "difference textures" $\mathcal{D}_i$ where $i \in \{1, ..., 4\}$ and weight maps $\mathcal{W}_{in} \in \mathbb{R}$. With these computed values, we efficiently approximate the $n$-th original texture $\mathcal{I}_n$ as

$$\mathcal{I}_n \simeq \bar{\mathcal{I}}_n = \mathcal{I}_0 + \sum_{i=1}^{4} \mathcal{D}_i \odot \mathcal{W}_{in}, \tag{1}$$

where the $\odot$ symbol stands for the Hadamart product, i.e., pixel-wise product. Figure 2 visually explains the reconstruction formulation in (1). Note that we have four source textures because the weight maps can be efficiently stored in the RGBA texture space. The weight $\mathcal{W}$ is smoothly defined, thus it can be stored in a downsampled image without significant loss of quality. In this paper, the ratio of down-sampling is eight, thus the size of the weight is 64 times smaller than the original one. With input texture size $W \times H$, the original textures require $N \times W \times H$ space, while the compressed model requires $(5 + N/64) \times W \times H$ resulting in a significantly smaller memory footprint especially when $N$ is large.

This paper presents a method to compute the decomposition of the input textures in (1) for the application of facial blendshape.

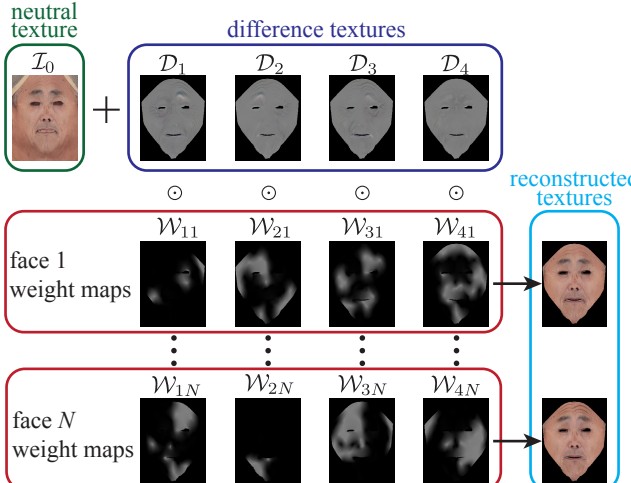

Figure 2: Texture-compression approach. For each blendshape mesh, four corresponding weight maps were obtained. These weight maps were then applied to the four difference textures. The summation of all weighted difference and neutral textures reconstruct the texture of the blendshape mesh.

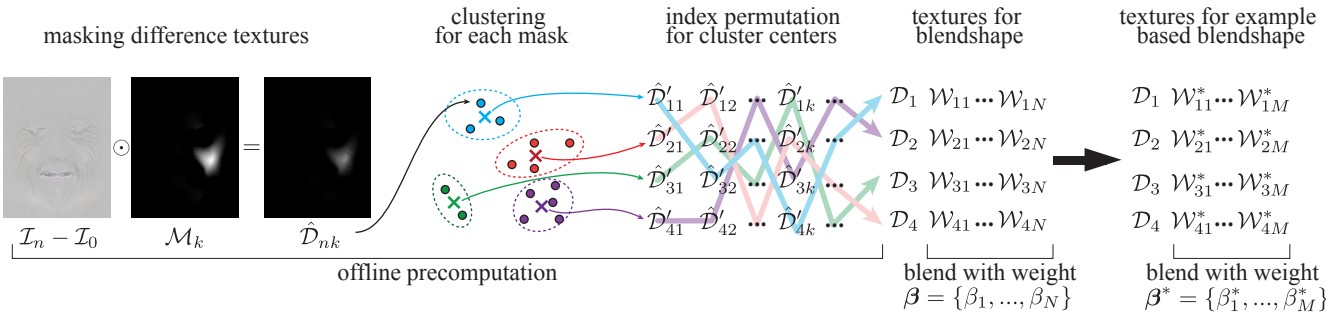

masking difference textures | clustering for each mask | index permutation for cluster centers | textures for blendshape | textures for example based blendshape

$\mathcal{I}_n - \mathcal{I}_0$    $\mathcal{M}_k$    $\hat{\mathcal{D}}_{nk}$

offline precomputation

blend with weight $\boldsymbol{\beta} = \{\beta_1, ..., \beta_N\}$

blend with weight $\boldsymbol{\beta}^* = \{\beta_1^*, ..., \beta_M^*\}$

Figure 3: Overview of our method. For each texture, we first compute the difference from the neutral texture and apply a mask to decompose the texture into localized fragments. Fragments from the same mask were split into four clusters. The fragments from different masks were reassembled into different textures using a combination that reduced the reconstruction error. In the runtime computation, the weight maps were blended with the blendshape weights. The weight map was augmented for the blendshape generated by example-based rigging.

Specifically, we generate textures for the example-based facial rigging [13], where the input textures are given in a small number of example shapes. When the weights of the blendshape change and thus the expression changes, our method outputs the texture with corresponding fine detail such as wrinkles.

We present an overview of our method in Figure 3. We first decompose the textures into fragments. Then, all the fragments are clustered into four groups. Finally, we find an optimal arrangement to store the fragments in the source textures and generate weight maps for every blendshape mesh. Section 3.1 explains this precomputation and Section 3.2 explains the following run-time computation.

### 3.1 Compression of Textures

The deformation of the entire face has rich varieties, but if we focus on a specific location, the deformation can be approximated with several modes. For example, a forehead exhibits horizontal furrows when the character raises the eyebrows and vertical furrows when the character frowns. To take advantage of such a locality, we first divide the input texture $\mathcal{I}_j$ into small fragments. This fragment can be simply computed by applying the mask $\mathcal{M}_k$ in the input texture

$$\hat{\mathcal{D}}_{nk} = (\mathcal{I}_n - \mathcal{I}_0) \odot \mathcal{M}_k, \tag{2}$$

where $k \in \{1, ..., \#\text{mask}\}$ is the index of the mask. Note that we apply the mask to the difference between the neutral texture $(\mathcal{I}_j - \mathcal{I}_0)$ here.

The mask takes the value between $[0, 1]$ where the value outside the fragment is zero. To make the seam less visible, the masks need to smoothly change their value in the 3D space. Moreover, we define the masks such that they add up to one $\sum \mathcal{M} = 1$.

**Mask Computation** We compute such smooth masks by solving the biharmonic equation (see Figure 4) inspired by the computation of the rigging weights in [10]. For the input mesh of the head, we first manually extract the set of triangles that corresponds to the face. Then we randomly sample the vertices of the extracted triangles. To sample uniformly over the 3D mesh, we use the Poisson disk sampling with the dart-throwing algorithm. Here, we reject samples within a $3\,\text{cm}$ radius. The number of the sampled vertices is the number of the masks $\#\text{mask}$. In total, we sampled 53 vertices.

Let $\phi \in \mathbb{R}$ are the values defined on the vertices of extracted face triangles. We solve the biharmonic equation by minimizing $||\Delta\phi||^2$ with the fixed boundary condition where $\phi = 1$ at one vertex and $\phi = 0$ at the other vertices. We use combinatorial Laplacian on the 3D mesh to robustly compute the minimization using an iterative solver for a sparse linear system. Finally, the $k$-th mask is computed by normalizing the solution in the UV space $\mathcal{M}_k = \phi_k / \sum_{k'=1}^{\#\text{mask}} \phi_{k'}$.

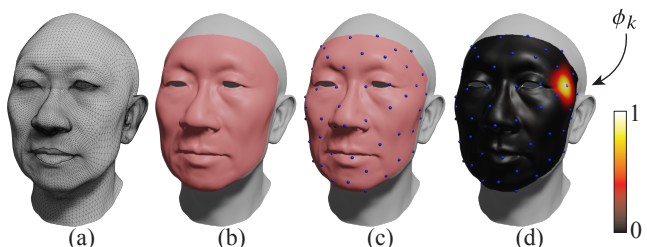

Figure 4: Mask computation approach. Given the input mesh (a), we first manually selected a region of interest (b). The vertices are then uniformly sampled over the selected region (c). Finally, the biharmonic equation is solved on the mesh while fixing the value at one of the sampled vertices as one and the others as zero (d).

**Clustering** The fragment of the input texture has several dominant modes. We extract such modes using K-means clustering method where the number of the cluster is four. Since the K-means clustering minimizes the variance inside the cluster, we can best select the four representative fragments of all fragments with the same mask. For each mask $k$, we record the four cluster centers of each cluster $\hat{\mathcal{D}}'_{ik}$ where $i \in \{1, ..., 4\}$.

**Difference Texture Generation** We now have four fragments for every mask, we are stitching these fragments back to four full textures. Naïvely adding up the fragments with the same texture index $i$ will lead to blurring the texture around the border of the fragments. Hence, we optimize the combination of the fragments for each texture such that the adjacent fragments match together. Let $\sigma(i, k)$ be a permutation of index $i$ for each mask $k$. We define the cost of a certain permutation index $\sigma$ as

$$C(\sigma) = \sum_{\{k_1, k_2\}=1}^{\#\text{mask}} \sum_{i=1}^{4} \left\| \hat{\mathcal{D}}''_{ik_1 k_2}(\sigma) - \hat{\mathcal{D}}''_{ik_2 k_1}(\sigma) \right\|^2, \tag{3}$$

$$\text{where} \quad \hat{\mathcal{D}}''_{ik_1 k_2}(\sigma) = \hat{\mathcal{D}}'_{\sigma(i,k_1)k_1} \odot \mathcal{M}_{k_2}. \tag{4}$$

Note that in (4), we apply the two masks $\mathcal{M}_{k_1} \odot \mathcal{M}_{k_2}$ to the difference texture (see the definition of the fragment in (2)). The Hadamard product of the two masks takes non-zero positive value only around the intersection of the mask $k_1$ and $k_2$.

The number of the possible permutation indexes $\sigma$ is finite but it is too large to compute by exhaustive search (i.e., $4!^{\#\text{mask}}$). Thus, we present a method to iteratively reduce the cost to find an approximate

minimizer. In each iteration, we look at a mask one by one. For each mask, we compute all the possible 4! index permutations for the mask and update the index to the minimizer while the indexes of other masks are fixed. By iterating this procedure 10 times, we reach convergence (see Figure 7-left). We repeat this procedure 100 times and record the permutation with the smallest cost.

Finally, we compute the four difference texture and the weights as

$$\mathcal{D}_i = \sum_{k=1}^{\#\text{mask}} \hat{\mathcal{D}}'_{\sigma(i,k)k}, \tag{5}$$

$$\mathcal{W}_{in} = \sum_{k=1}^{\#\text{mask}} \alpha_{ink} \mathcal{M}_k, \tag{6}$$

where $\alpha_{ink}$ takes the value 1 if the fragmented image $\hat{\mathcal{D}}_{nk}$ belongs to the $\sigma^{-1}(i,k)$-th cluster and otherwise it takes the value 0.

### 3.2 Texture Generation for Blendshapes

In this section, we describe texture generation for blendshape using our texture compression technique. The blendshape is often used to make a controllable face. Suppose we have a list of the vertices for the mesh of neutral face $V_0$ and the vertices for example face meshes $V_j$ the blendshape computes the vertex from the parameter $\beta_n \in [0, 1]$ where $n \in \{1, ..., N\}$ as

$$V(\boldsymbol{\beta}) = V_0 + \sum_{n=1}^{N} \beta_n (V_n - V_0) \tag{7}$$

Note that the blendshape formulation in (7) is specifically called *delta blendshape* [12] (see Figure 5-left).

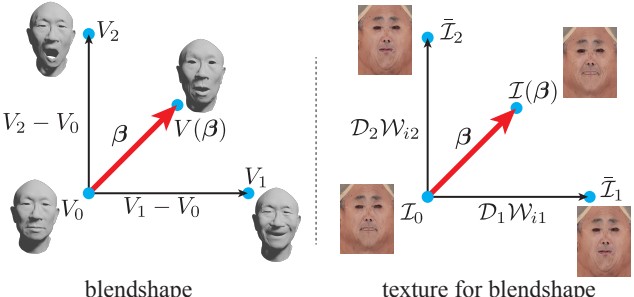

blendshape        texture for blendshape

Figure 5: We blend difference textures with the same weight as the blendshape weight.

With the textures that correspond neutral mesh $\mathcal{I}_0$ and the other example meshes $\mathcal{I}_n$, our compressed texture model compute the texture for the list of blendshape parameter $\boldsymbol{\beta}$ as

$$\mathcal{I}(\boldsymbol{\beta}) = \mathcal{I}_0 + \sum_{i=1}^{4} \mathcal{D}_i \odot \left( \sum_{n=1}^{N} \beta_n \mathcal{W}_{in} \right). \tag{8}$$

Note that in (8), we blend the texture with the ratio $\boldsymbol{\beta}$ in a similar manner as the blending for the vertex positions in (7) (see Figure 5). By blending the weight map $\mathcal{W}$, which is actually stored in the lower resolution, we can reduce the amount of computation.

Texture for Example-based Facial Rigging   We extend our texture compression for blendshape in (8) to the blendshape in example-based facial rigging in [13]. The example-based facial rigging generates blendshape meshes with new vertex positions $V_m^*$ where $m \in \{1, ..., M\}$ from the training meshes $V_n$ where

$n \in \{1, ..., N\}$. We aim to synthesize the texture for the generated blendshape $V_m^*$ when the texture $\mathcal{I}_n$ is given for the training meshes $V_n$. We first compute the weight map $\mathcal{W}_{in}$ and the difference texture $\mathcal{D}_i$ for the example meshes.

We refer to the original paper [13] for the detail of the example-based facial rigging. The basic idea of example-based facial rigging is to optimize the generated blendshape mesh $V_m^*$ such that the training mesh $V_n$ can be reconstructed as

$$V_n \simeq V_0 + \sum_{m=1}^{M} \bar{\beta}_{nm}(V_m^* - V_0), \tag{9}$$

where $\bar{\beta}_{nm} \in \mathbb{R}$ is the blending parameters that are optimized in the example-based facial rigging algorithm.

We synthesize the texture for generated blendshape such that it will reproduce the original compressed texture for the training meshes. This can be done by choosing the weight map for generated blendshape $\mathcal{W}_{im}^*$ to satisfy

$$\mathcal{W}_{in} = \sum_{m=1}^{M} \bar{\beta}_{nm} \mathcal{W}_{im}^*. \tag{10}$$

Unfortunately, the equation (10) alone cannot specify the weight map $\mathcal{W}_{im}^*$, since the equations are underdetermined (i.e., $N < M$). Our observation is that the weight maps should be sparse to avoid blending many weights. Hence, we minimize a regularizer $||\sum_{m=1}^{M} \mathcal{W}_{im}^*||^2$ to make the weight maps as small as possible while satisfying the constraint (10). Note that this regularization can result in negative values in the weight map. Since the weight will be applied to the difference textures the weight map is not required to be positive. The resulting weight map becomes

$$\mathcal{W}_{im}^* = \sum_{n=1}^{N} B_{mn}^+ \mathcal{W}_{in}, \tag{11}$$

where $B^+ = B^T(BB^T)^{-1}$ is the pseudo-inverse of the matrix $B = [\bar{\beta}_{nm}]_{N \times M}$. Finally, the texture for the example-based facial rigging is synthesized as

$$\mathcal{I}(\boldsymbol{\beta}^*) = \mathcal{I}_0 + \sum_{i=1}^{4} \mathcal{D}_i \odot \left( \sum_{m=1}^{M} \beta_m^* \mathcal{W}_{im}^* \right), \tag{12}$$

where $\boldsymbol{\beta}^*$ is the set of coefficients for example-based blendshape.

## 4 RESULTS

Evaluation Data   We evaluated our algorithm using high-resolution multi-view photos of 20 expressions purchased from a website [1]. For each expression model, we reconstructed a 3D mesh using the multiview stereo software, Metashape [2]. Because these models are not represented by a consistent mesh, we fit a base mesh to them using the commercial software Wrap [23]. The resulting mesh contains approximately 52k triangles.

In the supplementary video, we demonstrate our real-time dynamically changing texture for the blendshape implemented in Unity. All the weight maps were combined into one large RGBA texture, and a simple shader program blended these textures on the GPU in parallel. Hence, the cost of texture synthesis in(8) is negligible. For the example-based facial rigging [13], we use the implementation in the FaceScape [27] to generate the blendshape of 52 meshes that is compatible with Apple's ARKit.

**Performance**   Table 1 lists the computation times for offline precomputation and runtime memory consumption. The performance values were measured on a machine with Intel 12600 K and Windows 11 OS. The precomputation was implemented in Python. Specifically, we used the scikit-learn library for the k-means clustering.

For the example-based blendshape of 52 meshes, our method uses a similar amount of memory as the PCA baseline. Note that our method requires slightly more memory because we need to store the weight maps. Compared to the naïve approach of storing all the textures for the blendshape meshes our algorithm works on approximately 8 times smaller memory, which agrees with the estimation in Section 3.

**Texture Quality**   To evaluate our texture compression performance, we compared the root mean square error(RMSE) reconstruction errors against compression using PCA, which is often used in the texture compression of a blendshape [3]. For PCA, we choose the number of principal components to be four and eight with the mean texture to be the neutral texture. The four principal components take up roughly the same space as our methods. The RMSE error is computed as the average pixel difference in the RGB color values ranging from 0 to 255. The results of the comparison are illustrated in Figure 6. The error in the PCA is very large in areas such as the eyebrows and lips, where there are many fine details. For the ablation study, we also performed a comparison with compression without permutation optimization. We observed that our approach without permutation is generally better than the naïve approach of PCA. Our approach with weight maps and permutation optimization constantly produced the smallest errors compared to the other approaches.

**Choice of the Number of the Masks**   As we mentioned in Section 3, we choose four difference textures because the weight maps can be efficiently stored in the RGBA texture space, which is a popular texture format. The mask is used only in the precomputation, i.e., the run-time performance is not affected by the number of masks. Figure 7 shows the error statistics for different numbers of masks. We recorded the distribution of the RMSE reconstruction error for 1000 different random initializations. We observed that the global minimum decreased as #mask increased, However, if #mask is excessively large, it becomes difficult to reach the global minimum. For example, the maximum error increases and minimum error does not decrease in the case of for the #mask = 63. This is simply because finding the good permutation in the random search becomes more difficult compared to the case of the smaller number of masks. Generally, the global minimum can be reached within 20 tries when #mask = 5 but 1000 tries are not sufficient when #mask = 63. We chose 53 as the final choice of #mask because it maintains a good balance between reconstruction accuracy and complexity to find the overall minimum. Although our permutation optimization significantly reduces the error, proposing an efficient algorithm to find a near-optimal permutation will be one of our future works.

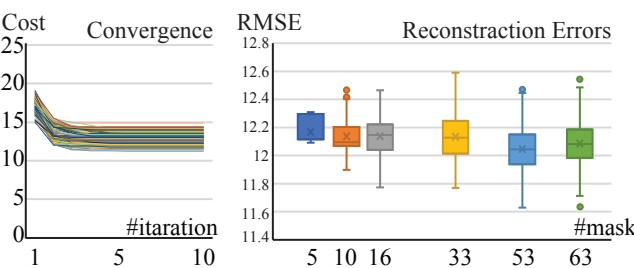

Figure 7: Left: Convergence of cost function for permutation optimization for 100 different initializations. Right: Box-and-whisker plots of the sum of the reconstruction errors when we change the number of masks and run permutation optimization with random initialization 1000 times.

**Textures for Example-based Blendshapes**   In Figure 8, we compare the quality of textures generated for the example-based blendshape (blending 52 meshes). For the baseline, we compute the texture for one of the example-based meshes using the weighted sum of the difference textures $\bar{\mathcal{I}}_m = \mathcal{I}_0 + \sum_{n=1}^{N} B_{mn}^+ (\mathcal{I}_n - \mathcal{I}_0)$. Here, we chose 6th blendshape (i.e., $m = 6$) of ARKit. Because the pixels were synthesized independently, the texture of the baseline model was full of noise. Moreover, unnatural wrinkles appeared because the baseline model did not use smooth weight maps. Our method achieves better results with significantly lower memory consumption, as shown in Table 1.

## 5   CONCLUSIONS AND FUTURE WORK

**Conclusion**   We present a method that can automatically compress the blendshape textures into one neutral texture and four difference textures. The different textures can then be summed up by spatially non-uniform weight maps that are smooth and stored at a low resolution. Thus, our method can significantly reduce memory consumption.

By combining high-resolution difference textures and low-resolution weight maps, the result of our method provides better localized details than global compression methods such as PCA. The smoothness of the weight maps also prevented artifacts from extreme values and noise when blending the weights in the blendshape model. Our method does not require any prior knowledge about the texture image, which is often required in data-driven methods, and is fully compatible with the blendshape for real-time applications.

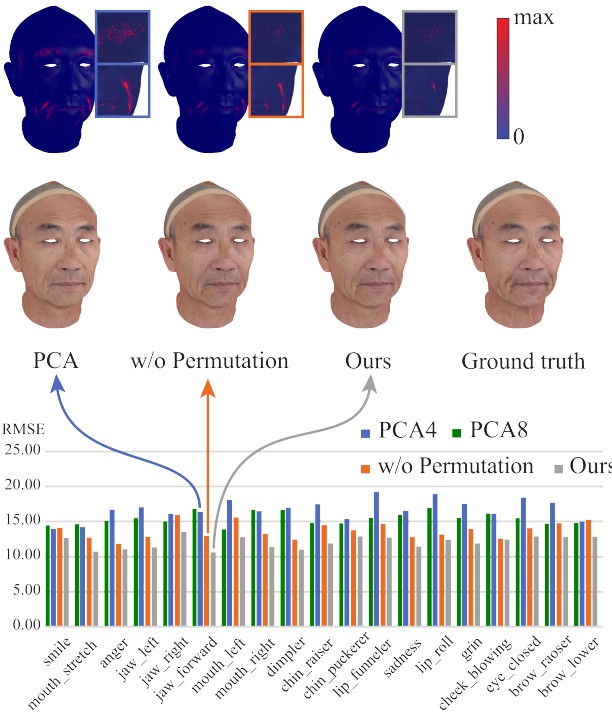

Figure 6: Reconstruction error of 19 blendshape textures computed by three different methods. Our method achieves the lowest error of three in all the texture reconstructions.

| Resolution | Computation Time (s) | | Memory Consumption (MB) | | | |
|---|---|---|---|---|---|---|
| | Clustering | Permutation | Ours | PCA(4) | PCA(8) | Raw data |
| $512 \times 512$ | 113 | 989 | 4.77 | 3.93 | 7.07 | 40.89 |
| $1024 \times 1024$ | 347 | 3447 | 19.07 | 15.72 | 28.30 | 163.58 |
| $2048 \times 2048$ | 1714 | 14502 | 76.28 | 62.91 | 113.24 | 654.31 |

Table 1: Timing of offline computation and the comparison of memory consumption for different methods measured for different texture resolutions for example-based blendshape (blending $53$ meshes). The "Clustering" stands for time for K-means clustering for all the fragments, "Permutation" stands for time to find the permutation with a small cost value.

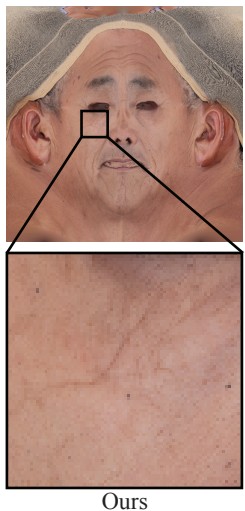 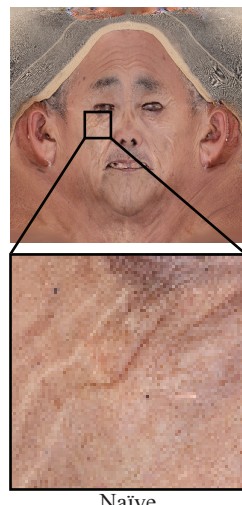

Ours                    Naïve

Figure 8: Reconstruction of texture on an example-based blendshape mesh. The naïve approach blends the example textures while our approach blends the weight maps. The naïve approach results in high-frequency noise and false wrinkles in white color.

**Limitation** The quality of our compression depends heavily on the alignments of the textures for all the example meshes. The compressed result will be blurry when blending miss-aligned textures. Besides, some artists would be more familiar with the traditional workflow where wrinkle textures are painted manually. Our current fully automatic workflow is not controllable by the users.

**Future Work** Currently, our work only compresses albedo textures, ignoring other textures such as normal, specular, and roughness maps. Although extending our compression method to other textures is straightforward, there is the potential to further compress the set of textures by leveraging the correlation between them. Another practical extension of this work is integrating the algorithm into the Unreal MetaHuman. We also intend to apply our method to facial textures. In particular, we were interested in efficiently representing wrinkles in garments in the texture space.

## ACKNOWLEDGEMENT

We thank the anonymous reviewers for their suggestions and comments. This work is supported by JSPS KAKENHI Grant Number 21K11910.

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
