# OpenReview forum: "FaceZip: Automatic Texture Compression for Facial Blendshapes"
_graphicsinterface.org/Graphics_Interface/2023/Conference_SD — GI 2023 - second deadline_

### Official Review · Reviewer_w78n · 2023-04-21
**need more limitation/discussion**

**Rating:** 7
**Confidence:** 4

**Review:**

The paper discusses the importance for realistic digital human models in various applications and the use of photogrammetry to capture 3D geometries and materials of actors. However, skilled artists are still needed to bring life to the captured model due to difficulties in parameterizing the model's surface material and deformation in a controllable way. The paper presents a technique to automatically generate compressed textures for blendshape models to represent details dynamically created with various expressions, using optimized selection of the combination of texture fragments while avoiding blur. The technique is demonstrated by comparing it to the baseline of principal component analysis and extending it to the blendshape generated by example-based blendshape technique.

A comparison is done between the proposed work and PCA-based work. As claimed in the introduction, PCA is a baseline proposed in the year of 1999. I wonder if there is any more recent stronger baseline to compare against.

Reading the paper, there appears to be a lack of discussion on the limitations or downsides of the proposed texture compression method. While the paper presents a thorough analysis of the advantages and benefits of the proposed method, it would be beneficial to have a more comprehensive examination of its potential disadvantages. Currently, the only downside mentioned is the slightly larger memory footprint, which is discussed in section 4. However, it is important to consider other aspects that could limit the use or effectiveness of the proposed method.

Regarding the proposed method's use of individual weight maps in the texture pipeline, it is unclear if there are any drawbacks associated with this approach. It would be useful for the authors to provide a detailed analysis of the benefits and limitations of using individual weight maps in texture synthesis. For example, do they require more computational resources or introduce any artifacts in the resulting textures? Does it introduce extra instability in regions with high curvatures or fine details? Such an analysis would help readers better understand the strengths and weaknesses of the proposed method and make informed decisions on its applicability in different scenarios.

---

### Official Review · Reviewer_6afZ · 2023-04-23
**The paper is interesting. Exposition can be improved.**

**Rating:** 6
**Confidence:** 4

**Review:**

This paper presents a method for compressing facial texture for blendshape facial animation. The method divides the face into 53 regions and selects four representative textures for each region, which are optimally combined to compose four facial textures. The proposed method achieves higher reconstruction accuracy compared to PCA-based compression, while maintaining a similar compression rate. Additionally, the paper proposes a method to generate a texture map suitable for example-based blendshape methods.

Although the proposed method is reasonable and effective, some design choices could benefit from further clarification. For example, the number of textures per region is set to four without explanation, and it would be useful to investigate how changing this number affects accuracy and compression rate.

Furthermore, the use of L2 norm for the regularization of the weights in Equation 11 may not guarantee non-negativity of beta. Additionally, if sparsity is desired as stated in the paper, L1 norm would have resulted in better sparsity. Lastly, in Figure 6, the reported RMSE ranges from 10 to 20, but it is unclear what this number represents.

Overall, the proposed method is a significant improvement over PCA-based compression for facial texture, but some additional explanations and clarifications would enhance the paper.

---

### Official Review · Reviewer_zAaY · 2023-04-24
**Interesting paper that solves a specific practical problem and could be of interest to the intended audience**

**Rating:** 7
**Confidence:** 3

**Review:**

Interesting paper that solves a specific practical problem and could be of interest to the intended audience

This paper presents a method for automatically compressing the textures associated with facial blend shapes. The method is sufficiently explained for a 6 page paper.

Pros
1. The proposed method offers memory reduction comparable to the standard PCA method.
2. The proposed method produces results that have lower RMSE error compared to PCA.
3.  It does not seem to have unreasonable constraints or restrictions on the blend shapes or textures.
4.  The way masks are used is simple but effective.

Cons
1. The proposed method works currently only on the albedo textures and not on normal maps, roughness maps etc.
2. The precomputation steps, although described, can be tedious, but that is not a fault the method.
3. The paper does not mention the number of principal components that the baseline method uses or how they were chosen. Since this is the main point of comparison more details are needed. For example, could PCA trade off some memory savings for improved performance? After all it seems to use less storage than the proposed method.

I would have liked to see more data points in Figure 7 (right). Going from 16 to 33 is a big jump in the number of masks. I wonder if the method could have been tested on the Unreal Engine Metahumans.

The video is interesting but I have liked to see more comparisons and results. For example, it would be interesting to see the biggest differences in the results in relation to the number of masks, and other parameters of the method.

I think that overall the paper is of interest to the audience of GI.